# The Mantle Transcriptome of *Chamelea gallina* (Mollusca: Bivalvia) and Shell Biomineralization

**DOI:** 10.3390/ani12091196

**Published:** 2022-05-06

**Authors:** Federica Carducci, Maria Assunta Biscotti, Alessandro Mosca, Samuele Greco, Marco Gerdol, Francesco Memmola, Marco Barucca, Adriana Canapa

**Affiliations:** 1Dipartimento di Scienze della Vita e dell’Ambiente, Università Politecnica delle Marche, Via Brecce Bianche, 60131 Ancona, Italy; f.carducci@univpm.it (F.C.); moxale8@hotmail.it (A.M.); f.memmola@univpm.it (F.M.); m.barucca@univpm.it (M.B.); a.canapa@univpm.it (A.C.); 2Dipartimento di Scienze della Vita, Università degli Studi di Trieste, Via L. Giorgieri 5, 34127 Trieste, Italy; samuele.greco@phd.units.it (S.G.); mgerdol@units.it (M.G.)

**Keywords:** mollusc, biomineralization, gene expression analysis, transcriptomics

## Abstract

**Simple Summary:**

*Chamelea gallina* is a bivalve mollusc that represents one of the most important fishery resources in the Mediterranean basin. In this species, the thickness and sturdiness of the shell valves are two important characteristics as they are decisive for protection against predators and survival of specimens rejected in the sea because caught under commercial size. The aim of this work is to investigate the ability of this species to modulate the expression of genes encoding proteins involved in shell biomineralization process in response to abiotic and biotic factors. Our findings, obtained through a multidisciplinary approach, highlighted a different shell mineralization behaviour in *C. gallina* clams collected in sampling sites characterized by different salinity and food availability. Moreover, this study provided the first comprehensive transcriptome from mantle, the tissue responsible for shell formation. Therefore, these results contribute to increase knowledge on this process and might help in adopting ad hoc management plans for this fishery resource.

**Abstract:**

The striped venus *Chamelea gallina* is a bivalve mollusc that represents one of the most important fishery resources of the Adriatic Sea. In this work, we investigated for the first time the ability of this species to modulate the expression of genes encoding proteins involved in biomineralization process in response to biotic and abiotic factors. We provided the first comprehensive transcriptome from the mantle tissue of clams collected in two sampling sites located along the Italian Adriatic coast and characterized by different environmental features. Moreover, the assessment of environmental parameters, scanning electron microscopy (SEM), and X-ray diffraction (XRD) measurements on valves were conducted to better contextualize RNA sequencing (RNA-Seq) data. Functional annotation of differentially expressed genes (DEGs) and SEM observations highlighted a different shell mineralization behaviour in *C. gallina* clams collected from two selected sites characterized by diverse environmental parameters.

## 1. Introduction

Bivalves evolved a biomineralized shell as strategy to protect against predators, support their soft-bodies, and store minerals [1,2]. In venerids, this structure presents the periostracum, the outermost thin organic leathery layer, a middle layer composed of two sub-layers (the inner one that is homogeneous and the outer one that present crossed-lamellar structure), and the inner layer formed of irregular granules or crossed-acicular structure. However, differences can be observed in various genera, even at single species level [3]. In general, in molluscs, the major constituent of shell is represented by calcium carbonate, one of the most abundant minerals in nature, inserted in an organic protein matrix that rarely exceeds 5% of the total shell weight [4]. Although the mechanism of shell formation is not yet fully understood, it has been reported that the epithelial mantle cells are able to concentrate amorphous calcium carbonate and release it. The shell matrix proteins (SMPs) are involved in the deposition of carbonate calcium crystals influencing the shell shapes [5].

Proteomic, transcriptomic, and genomic analyses have allowed the identification of SMPs, characterized by the presence of functional domains such as von Willebrand factor type A (VWA), chitin-binding (CB), carbonic anhydrase (CA), and acidic domains [6,7,8]. In addition, transgelin-like and calponin proteins, which both contain the calponin homology (CH) domain, have been proposed to be involved in biomineralization in several bivalve species [9]. In particular, calponin proteins stabilize the SMPs [10]. Moreover, recent literature available on bivalve biomineralization has evidenced that tyrosinases are involved in shell matrix and periostracum formation [10,11].

The bivalve *Chamelea gallina* (common name: striped venus) is one of the most important commercial fishery resources in the Mediterranean Sea [12], with catches mostly occurring through the use of dredges and bottom trawls. Biomineralization is deeply influenced by environmental factors, Gizzi and colleagues (2016) [13] have evaluated the effect of solar radiation and temperature on the shell properties of *C. gallina* clams of commercial size. The authors have reported a higher porosity in the shell of clams living in warmer areas without evidencing any changes in its composition and microstructure. Later, in addition to the aforementioned environmental parameters, Mancuso and co-workers (2019) [14] have also considered the impact of sea surface salinity and chlorophyll concentration on shells collected from immature and mature clams evidencing an influence of these factors on calcification. However, the ability of this species to modulate the expression of genes encoding proteins involved in biomineralization in response to these biotic and abiotic factors still remains unexplored. The recent study published by our group on the striped venus using an omics approach has showed that this species is able to modulate gametogenesis in response to the energy resources available. Indeed, differences in gene expression linked to this biological process and gonad maturation emerged between clams collected in two sampling sites located along the Italian Adriatic coast and characterised by different food availability [12].

The biomineralization process influences thickness and sturdiness of the shell valves, two important characteristics decisive for protection against predators and survival of *C. gallina* specimens rejected in the sea when caught under commercial size. Therefore, the aim of this work was to evaluate whether the transcriptional activity of genes involved in biomineralization process in *C. gallina* clams collected from two selected sites characterized by different environmental parameters. We provide for the first time a comprehensive transcriptome from the mantle tissue of clams collected in the two selected sampling sites located along the Italian Adriatic coast and the functional annotation of the differentially expressed genes (DEGs) related to biomineralization process. To better contextualize RNA sequencing (RNA-Seq) data, an assessment of environmental parameters was performed. Moreover, scanning electron microscopy (SEM) and X-ray diffraction (XRD) measurements were conducted in order to evaluate any potential change in the shell matrix deposition.

## 2. Material and Methods

Organisms used in this study are invertebrates (molluscs) that do not need approval from the ethics committee. *C. gallina* clams of similar size (26.9 ± 1.1 mm shell length) were collected by hydraulic dredges in two different sites, Senigallia (S) 43°37′38″ N/13°26′28″ E (Marche region) and Silvi Marina (SM) 42°48′51″ N/13°57′41″ E (Abruzzo region), in May 2020 (Figure 1).

### 2.1. Mantle Transcriptome Sequencing and Assembly

For each sampling site, three RNA pools (each constituted by RNA extracted from mantle tissue of 10 specimens of the same size) were analysed as replicates. Total RNA was extracted from mantle tissue using Trizol^®^ reagent (Invitrogen, Thermo Fisher Scientific, Waltham, MA, USA) and quantity and purity were checked using Qubit 4.0 fluorometer and NanoDrop (Thermo Fisher Scientific), respectively. The six RNA pools were sent to CD Genomics (Shirley, NY, USA; https://www.cd-genomics.com) where RNA quantity was verified using Ribogreen method using Victor X2 fluorometer and the RNA integrity number (R.I.N.) evaluated by Agilent RNA Screen Tape (Agilent technologies, Santa Clara, CA, USA). TruSeq Stranded mRNA Sample Prep kit (Illumina, San Diego, CA, USA) was used for library preparation following the manufacturer’s instructions, using 1 µg of good quality RNA (R.I.N. > 7) as input. The fragmentation of the poly-A mRNA was performed by heath exposure at 94 °C for 3 min and every purification step was carried out using 1× Agencourt AMPure XP beads. Illumina cBot for cluster generation on the flowcell was used to process libraries, following the manufacturer’s instructions and sequenced with a 2 × 151 bp paired-end mode on NovaSeq 6000 platform (Illumina, San Diego, CA, USA).

Adapter trimming was conducted using fastp v0.20.1 [15] with the following parameters: -f 15 -F 15 -g -x -5 -3 -n 0 -l 175. Trimmed reads obtained from the six pools, deposited in the NCBI SRA database under the BioProject xx PRJNA, were used as an input for de novo assembly with Trinity v2.12 [16] using default parameters and a minimum allowed contig length of 200 nucleotides.

The assembled transcriptome was filtered with the “get_longest_isoform_seq_per_trinity_gene.pl” script provided with Trinity, this step allowed the removal of all the shorter isoforms derived from alternative splicing, resulting in a non-redundant assembly. The overall completeness of the reference transcriptome assembly was evaluated with BUSCO v5.0.0 [17] which also allowed measurement of duplication and fragmentation rates, with reference to the Mollusca OrthoDB v10 database [18]. Functional annotation was carried out with Trinotate v3.2.1 [19]. All contigs were translated to putative protein sequences using TransDecoder v3.0.1 (https://github.com/TransDecoder/TransDecoder/releases) and annotated based on significant BLASTX and BLASTP [20] matches in the UniProtKB sequence database [21]. These homologies were used to associate each contig to cell component, molecular function, and biological process gene ontology terms [22]. Protein sequences were also analysed with Hmmer v3.2.1 [23], searching for conserved domains included in the Pfam v31.0 database [24].

### 2.2. Transcriptome Refinement

Abundance estimates of reads were first calculated by mapping the reads of each pool to the *C. gallina* non-redundant reference transcriptome using QIAGEN CLC Genomics Workbench v12.0 (https://digitalinsights.qiagen.com/), setting the length and similarity fraction parameters to 0.75 and 0.98, respectively. Gene expression levels were computed as transcripts per million (TPM) [25]. Based on read mapping distribution, a gene expression threshold value (TPM = 10) was set in order to remove possible exogenous contaminations and short contigs with low coverage, most likely resulting from the fragmentation of transcripts expressed at low levels or from pervasive intergenic transcription. All the contigs that did not reach this threshold in either of the two analysed samples were discarded. The suitability of such TPM threshold was checked by performing another completeness evaluation using BUSCO v5.0.0 [17] as reference the Mollusca OrthoDB v10 database [18].

Since the resulting transcriptome still contained contigs derived from transcripts encoded by the mitochondrial genome and ribosomal RNAs, it was further filtered by running BLASTN against the reference sequences available in the NCBI nucleotide database (EU660751.1, DQ458474.1). Significant hits (*e*-value < 1 × 10^−5^) were removed, creating the reference transcriptome used for the following gene expression analyses.

### 2.3. Gene Expression Analyses

CLC genomics workbench v12.0 (https://digitalinsights.qiagen.com/) was used to map the trimmed reads to the reference transcriptome with the RNAseq analysis tool (length fraction = 0.75, similarity fraction = 0.98) producing Gene Expression tracks for all samples, which were then forwarded to the Differential Expression in Two Groups tool. Such tool implements differential gene expression analysis using a Generalized Linear Model (GLM). Differentially expressed genes (DEGs) were determined for the pairwise comparison considering S as the test and SM as the control and by applying thresholds for false discovery rate (FDR) corrected *p*-value < 0.05 and absolute value of fold change >4.

### 2.4. DEGs Functional Enrichment Analysis

Gene ontology (GO) [22] terms derived from the functional annotation of the transcriptome were used for a hypergeometric test-based enrichment analysis of DEGs. The resulting enriched GO terms were filtered for FDR corrected *p*-value < 0.05 and observed–expected values > 3 in order to remove false positive and non-informative hits. Graphical representation of the results from the enrichment analysis was achieved with the GOplot [26] R package.

### 2.5. X-ray Diffraction (XRD) and Scanning Electron Microscopy (SEM) Measurements

For each sampling site, X-ray diffraction (XRD) experiments were performed on a subset of specimens from those selected for RNA-Seq analyses. XRD patterns were recorded on a D8 Advance X-ray diffractometer (D8 Bruker-AXS, Madison, WI, USA) using Ni-filtered CuKα monochromatized radiation at 40 kV, 40 mA, and 25 °C. The 2θ range from 20° to 60° was covered at a step size of 0.02°/4 s. The spacing of Bragg peaks detected were reported within the same 2θ range and peak indexing was determined after a calibration made using a quartz standard. 

For each sampling site, SEM observations were carried out on a subset of individuals from those selected for RNA-Seq analyses to check any possible variation due to different environmental conditions on micro-textural characteristics of *C. gallina* shell. After being resin mounted, samples were investigated performing a transversal valve section. Each section was etched with an acetic acid solution (1% *v*/*v*) for 1 min to remove debris and artefacts from cutting. Samples were coated with a gold layer (5 nm) and analysed with a TESCAN VEGA3 at 10 kV of beam intensity and images were acquired up to 5000× of magnification. 

### 2.6. Assessment of Environmental Parameters

The analysis concerned daily means for temperature, salinity, and chlorophyll in the two sampling stations S and SM and spanned the year starting from September 2019 to September 2020. The dataset was extracted from the historical timeseries from the Mediterranean Sea Physics Analysis and Forecast dataset [27] for temperature and salinity while the Mediterranean Sea Biogeochemistry Analysis and Forecast dataset was used to extract chlorophyll timeseries [28]. The two datasets were taken from the output of hydrodynamics and biogeochemical models of the Mediterranean Sea, publicly available on E.U. Copernicus Marine Service Information (https://marine.copernicus.eu accessed on 15 March 2021). The two model grid points closed to the sampling stations of our interest were chosen and the data at the sea-floor level were extracted. Monthly and seasonal means of temperature, salinity, and chlorophyll over the time span of interest were calculated. Moreover, these data were statistically processed in order to find significant difference between stations and seasons through the analysis of variance (ANOVA) followed by a Tukey’s test as post hoc pairwise comparison. Since the assumptions of the homogeneity of variances (tested with the Cochran’s C-test) were not verified, the level of significance was set at 0.01. The experimental design included two factors: station (random, with two levels “S” and “SM”) and season (fixed, with four levels “winter”, “spring”, “summer”, and “autumn”).

## 3. Results

### 3.1. Transcriptome Assembly and Annotation

For each sampling sites (Senigallia, S and Silvi Marina, SM), three RNA pools (each constituted by RNA extracted from mantle tissue of 10 specimens of the same size) were analysed as replicates. The RNA sequencing yielded a total of 307,571,570 raw nucleotide paired-end reads. After adapter trimming and filtering based on quality and length, a total of 288,550,786 paired-end reads were retained and used for the de novo assembly. The non-redundant transcriptome assembly obtained from the six pools comprised 462,493 contigs whose size ranged from 201 to 30,079 bp, with an average length of 436.27 bp and a N50 length of 452 bp (Appendix A). The de novo assembled transcriptome displayed a good level of completeness with the 77.2% of complete and single-copy (S), 7.9% of complete and duplicated (D), 3.6% of fragmented (F), and the 11.3% of missing (M) Mollusca BUSCOs (Appendix A). After the filtering steps, the transcriptome consisted of 63,039 sequences, with average contig length of 1094.12 bp and N50 length of 1689. The BUSCO scores of the reference transcriptome are as follows: 77.2% S, 4.3% D, 4.2% F, and 14.3% M. In total, 40,774 contigs were annotated by either blast (against Uniprot) or Hmmer (against Pfam). Of these, 25,446 sequences were annotated with gene ontology terms.

### 3.2. Gene Expression Analyses

A total of 364 DEGs were obtained from the S vs. SM comparison, including 95 upregulated and 269 downregulated. Among the 95 upregulated DEGs, only the 13.68% were annotated with PFAM hits and a further 8.4% and 6.3% displayed BLASTP or BLASTX hits, respectively. On the other hand, 35.3% of the downregulated genes showed a PFAM hit, with an additional 10% and 1.5% showing BLASTP and BLASTX hits, respectively. The complete list of statistically significant DEGs with related annotations was provided in Appendix A.

Only three entries related with biomineralization process were identified among the upregulated DEGs for samples of the S sampling site based on PFAM annotation. These genes encode proteins containing the “Common central domain of tyrosinase”, the “Kunitz/Bovine pancreatic trypsin inhibitor (BPTI) domain”, and the “EF-hand domain pair”. The first one, displaying a fold change (fc) of 14.6, was among the top five upregulated genes; the second one showed an fc of 5.9; the latter an fc of 5. Concerning the downregulated genes, a total of five entries encoded proteins containing domains known to be part of the biomineralization toolkit: two include the “Kazal-type serine protease inhibitor domain” (fc of −15.4), and the other three include the “Common central domain of tyrosinase” (fc of −4.6), the “EF-hand” (fc of −4.5), and the “Eukaryotic-type carbonic anhydrase” (fc of −4.2). Moreover, other downregulated DEGs correlated with biomineralization were: “Ubiquitin”, “tubulin” (three entries), “spectrin”, “phosphoenolpyruvate kinase” (PEPCK) (16 entries), “calponin family repeat”, “calcium-binding EGF domain”, and “cation transport ATPase (P-type)”. The last three entries are proteins involved in calcium transport (Appendix A).

The manual inspection of BLASTP and BLASTX searches allowed us to group the upregulated DEGs within two relevant macrocategories in the context of shell formation: *biomineralization related proteins* (5 entries) and *microtubule associated proteins* (1 entry). Downregulated genes could be grouped in four macrocategories: *biomineralization related proteins* (16 entries), *metal ion binding* (14 entries), *energy metabolism* (27 entries), *microtubule associated proteins* (10 entries), and *protein trafficking* (1 entry) (Table 1).

Although only part of the DEGs were associated with an annotation, 354 (288 biological process, 63 molecular function, and 3 cellular component) GO terms were significantly enriched in this gene subset (Figure 2, Appendix A). The associated biological processes mostly involved *digestion*, *catabolism*, *metabolism*, *cellular response to external stressors*, *immune response*, and *reproduction*. Molecular functions involved several *molecule bindings*, *oxidoreductase activity*, *DNA binding/synthesis*, and *phosphorylation*. The only three significantly enriched cellular components were *mitochondrion*, *extracellular region*, and *endoplasmic reticulum*.

### 3.3. X-ray Diffraction (XRD) and Scanning Electron Microscopy (SEM) Measurements

X-ray diffraction (XRD) profiles evidenced a shell mineral composition made entirely by aragonite for clams sampled from both sampling sites (Appendix A). On the contrary, variations in the microstructure of the shells of clams considered in this study emerged from scanning electron microscopy (SEM) measurements. Figure 3 provides a summary of SEM images at 500× of magnification. 

Overall, clams sampled at the S site displayed a more homogeneous organization of the shell matrix. The inhomogeneity emerged from the SEM images obtained for individuals of SM sampling site was attributable to a higher porosity of their outer shell prismatic layer. Moreover, differences in the transition layer of the shell emerged from higher SEM magnification at 1000×, 3000×, and 5000×, suggesting a possible different organization of aragonite crystals in the outer prismatic layer between clams collected from the two sampling sites.

### 3.4. Assessment of Environmental Parameters

The extraction of timeseries was made for twelve months, spanning from September 2019 to September 2020. The timeseries of the daily means of temperature, salinity, and chlorophyll are shown in Figure 4, while the monthly and seasonal averages and standard deviations for the two sampling stations are shown in Appendix A. 

No significant difference (*p* value > 0.01; see Appendix A) was found between the two considered stations for temperature. This parameter displayed a remarkable seasonal variation with significant difference within each station (*p* value < 0.01; see Appendix A for the ANOVA results and Appendix A for the results of the Tukey’s test pairwise comparisons among seasons). The only exception was represented by the comparison between Autumn and Spring for both stations. The salinity, with few exceptions (see Appendix A), was also characterized by a significant seasonal pattern in each station. However, in this case, significant difference between the two sampling stations was also found in terms of seasonal salinity (Appendix A) for the analysed timeseries. The salinity was always lower at the S station. The chlorophyll maximum was reached during the winter with S station characterized by a higher productivity (Figure 4).

## 4. Discussion

The bivalve *Chamelea gallina* is one of the most important commercial fishery resources [12] in the Adriatic Sea, with annual catches for about 15,000 tonnes (for the year 2018). The use of dredges and bottom trawls can determine the breakage of shells of these animals [29,30] with negative consequences on fishery yields and survival of undersized clams returned to the sea after sorting. Therefore, to counteract these unfavorable aspects, increasing attention has been focused on the study of biomineralization [13], since this process influences thickness and sturdiness of the shell valves.

In this study, for the first time, a transcriptomic approach was used to investigate the expression of genes related to shell formation in specimens of *C. gallina* sampled from two sites located along the Italian Adriatic coast. These data were coupled with electron scanning microscopy (SEM) and X-ray diffraction (XRD) measurements and contextualized with the assessment of environmental parameters.

The comprehensive de novo transcriptome assembly from the mantle of *C. gallina* specimens showed comparable quality with the recently produced transcriptomes from the same tissue of *Mytilus edulis* [31,32]. Moreover, BUSCO assessment evidenced a high level of completeness with the detection of 77.2% of complete Mollusca BUSCOs.

The comparison between the two sampling sites, S and SM, showed differences in the expression levels of genes involved in the biomineralization process. Within the S upregulated genes, the DEGs functional annotation revealed the presence of “Common central domain of tyrosinase” that was found to be involved in the formation of periostracum [33,34] and recently reported to be upregulated during shell repair processes [32] and hardening [11]. The “Kunitz/BPTI domain” was generally found in proteins having a proteinase inhibitor role, shown to be directly involved in the nucleation and/or growth and termination of crystal calcification [32,35]. Jin, Liu, and Li (2019) [36] have reported a role of Kunitz protease inhibitor in crystal overgrowth due to an increased calcium carbonate precipitation. The third upregulated domain found in S sampling site was “EF-hand”, which is considered a SMP-associated domain 8.18 involved in calcium-binding during shell formation [37,38]. 

The presence of few genes encoding proteins containing the domains “Common central domain of tyrosinase” and the “EF-hand” also among down-regulated DEGs was not a surprising finding, considering the role played by these domains in shell formation [8,11,32,39]. Among the downregulated genes, the presence of “Eukaryotic-type carbonic anhydrase” and the “Kazal-type serine protease inhibitor” domains were also interesting. The former is a SMP known to be part of the bivalve biomineralization toolkit [6,8,38,40] while the latter plays a role similar to that of “Kunitz/BPTI domain”. 

The analysis of the functional annotations associated with these DEGs suggested that clams sampled in the same period (May 2020) and having the same dimensions were in a less active biomineralization process at the S sampling site compared with the SM sampling site. These locations belong to two different biogeographic zones [41] placed in the north and south of Monte Conero, respectively. Water circulation and the runoff waters coming from the Po River (the main northern Adriatic River) are changed by this promontory, leading to a lower salinity at S sampling site. This is in line with observation reported by Mancuso and colleagues in 2019 [14], in which the authors described a higher net calcification rate in clams living in areas characterized by a higher salinity level. Moreover, for the year considered in the present work, the assessment of environmental parameters showed a higher food availability for the S sampling site that might be interpreted with a major amount of energy to be addressed to reproduction, probably responsible for a delayed growth. On the contrary, clams of the SM sampling site, having a lower quantity of food available, would undertake gametogenesis later, directing the available energy resources towards growth and biomineralization. Our data did not suggest an influence of temperature since this parameter was constant in the two sampling sites.

This was also in agreement with functional enrichment analyses performed in this study showing a less active energy metabolism in S clam mantle. Moreover, SEM observations showed a more compact shell for specimens of the S sampling site contrarily from those of SM characterized by a higher porosity. The differences in shell microstructure can be also influenced by changes in salinity evidenced between the two sampling sites due to the Po River freshwater contribution [14,42].

These findings suggested that the low number of domains identified among the S upregulated DEGs belongs to genes probably involved in the shell hardening, while those found among the SM upregulated DEGs (i.e., downregulated genes of the S sampling site) might be related with an ongoing biomineralization. 

In conclusion, in this work we provided a first comprehensive mantle transcriptome of *C. gallina* that allowed also in this species the characterization of domains already known to be involved in biomineralization process. The characterization of DEGs highlighted a different behaviour during shell mineralization in clams collected from the two selected sampling sites having different environmental features. These findings, contextualized through a multidisciplinary approach, suggest that the food availability and salinity might influence shell formation in *C. gallina*. Overall, gathering an improved knowledge on this process might help in adopting ad hoc management plans for this fishery resource.

## Figures and Tables

**Figure 1 animals-12-01196-f001:**
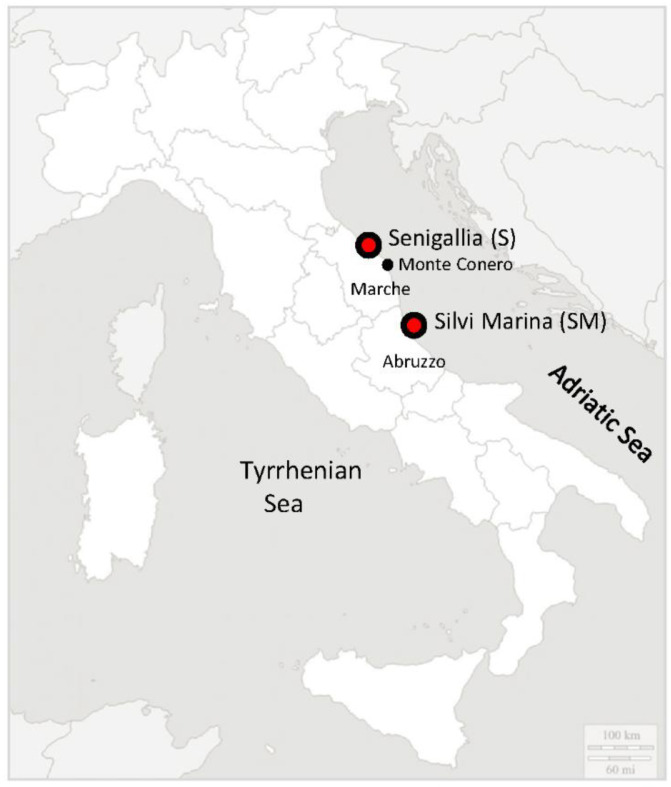
Geographical localization of sampling sites considered in the present work.

**Figure 2 animals-12-01196-f002:**
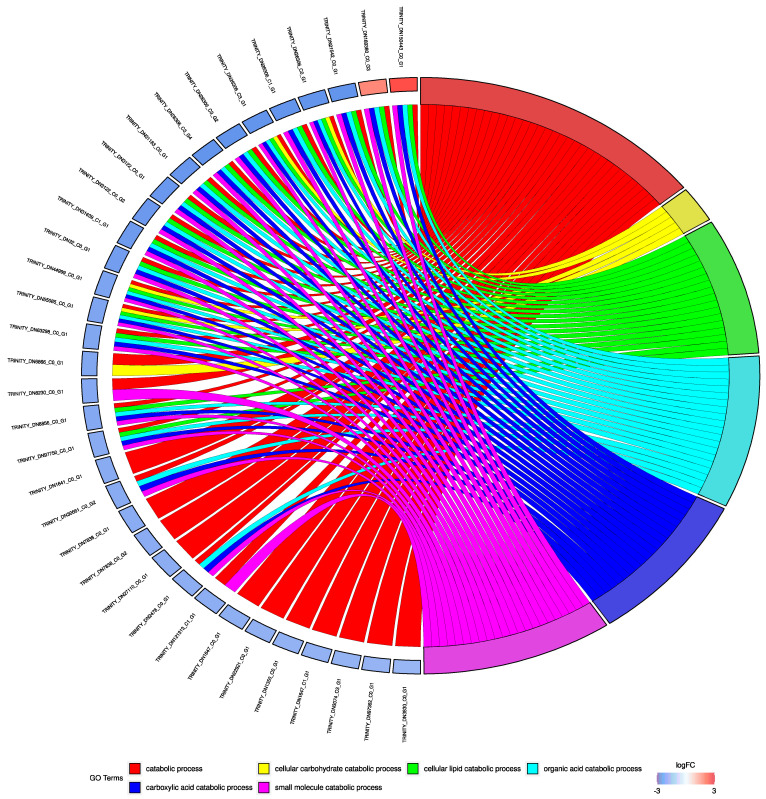
Circos plot obtained from functional enrichment analysis performed on differentially expressed genes derived from the comparison S vs. SM.

**Figure 3 animals-12-01196-f003:**
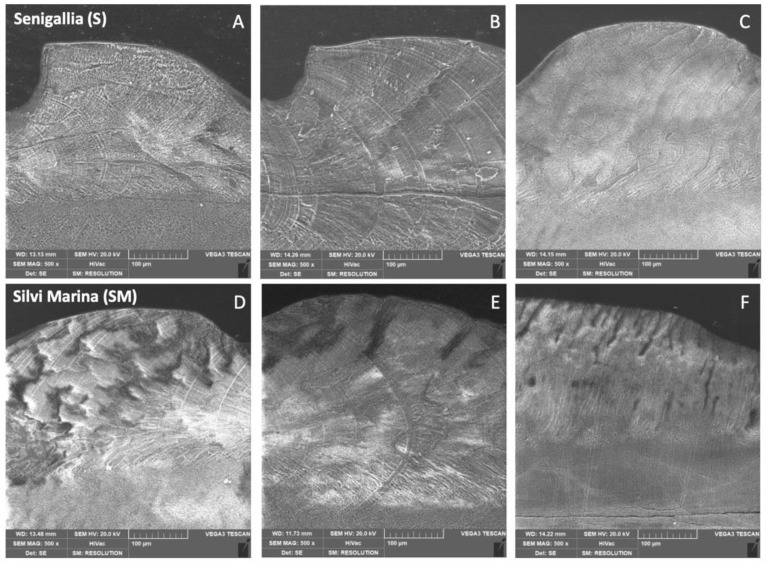
Scanning electron microscopy (SEM) observations obtained at 500× of magnification. (**A**–**C**) are referred to specimens of the Senigallia (S) sampling site. (**D**–**F**) are referred to specimens of the Silvi Marina (SM) sampling site.

**Figure 4 animals-12-01196-f004:**
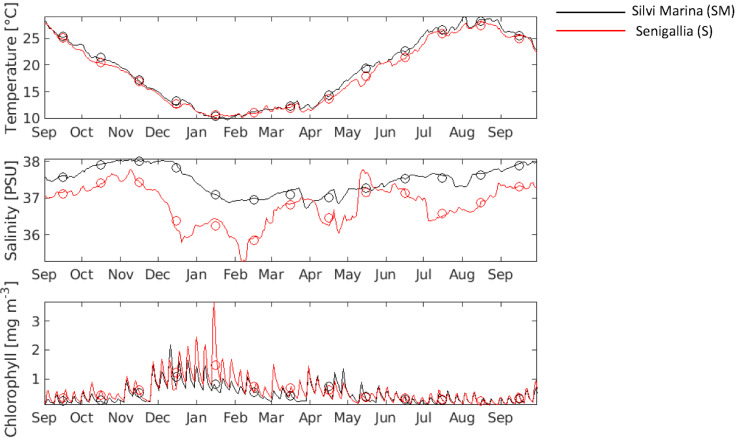
Timeseries of environmental parameters. Daily temperature, salinity, and chlorophyll at the sampling stations of Senigallia (S, red line) and Silvi Marina (SM, black line) were reported. Computed monthly means are superimposed as circles.

**Table 1 animals-12-01196-t001:** Differentially expressed genes (DEGs) obtained for Senigallia (S) and Silvi Marina (SM) sampling sites and their related annotation description and entries found.

**Upregulated at S Sampling Site**	**BLAST Results**	
macrocategories	description_UNIPROT accession	entries found
biomineralization related proteins	Putative tyrosinase-like protein tyr-3 (TYR3_CAEEL)	1
	WAP, Kazal, immunoglobulin, Kunitz and NTR domain-containing protein 2 (WFKN2_MOUSE)	1
	Sarcoplasmic calcium-binding protein (SCP_MIZYE)	1
	Early growth response 1 (EGR1_XENTR)	2
microtubule associated proteins	Janus kinase and microtubule-interacting protein 3 (JKIP3_HUMAN)	1
**Downregulated at S Sampling Site**	**BLAST Results**	
macrocategories	description_UNIPROT accession	entries found
biomineralization related proteins	insoluble shell matrix protein 6 (IMSP6_RUDPH)	6
	insoluble shell matrix protein 3 (IMSP3_RUDPH)	3
	insoluble shell matrix protein 2 (IMSP2_RUDPH)	1
	ubiquitin (UBIQ_LUMTE)	1
	mucin-like protein (MLP_ACRMI)	1
	tyrosinase-like protein (TYRO_PINMA)	1
	perlucin (PLC_HALLA)	1
	galaxin (GXN_ACRMI)	1
	carbonic anhydrase 7 (CAH7_HUMAN)	1
metal ion binding	protocadherin FAT4 (FAT4_MOUSE)	1
	calcium binding and coiled coil domain-containing protein 2 (CACO2_MACFA)	1
	CBL-interacting serine-threonine protein kinase 5 (CIPK5_ARATH)	1
	cAMP and cAMP-inhibited cGMP 3′,5′-cyclic phosphodiesterase 10A (PDE10_HUMAN)	1
	Zinc finger MYM-type protein 4 (ZMYM4_MOUSE)	1
	baculoviral IAP repeat containing protein 7 (BIRC7_MOUSE, BIRC_XENTR)	2
	DNA-directed RNA polymerase II subunit RPB1 (RPB1_PLAFD)	1
	sodium/potassium-transporting ATPase subunit alpha-B (AT1B_ARTSF)	1
	thyroid peroxidase (PERT_MOUSE)	1
	cysteine dioxygenase type 1 (CDO1_BOVIN)	1
	NADPH oxidase 5 (NOX5_HUMAN)	1
	metal cation symporter ZIP14 (S39AE_BOVIN)	1
	sodium/potassium-transporting ATPase subunit alpha (AT1A_TAESO)	1
energy metabolism	phosphoenolpyruvate carboxykinase cytoplasmic (PCKCG_BOVIN, PCKG_DROME, PCKGM_MOUSE, PCKCG_CHICK)	18
	Endoglucanase E-4 (GUN4_THEFU)	1
	Isocitrate dehydrogenase [NADP] cytoplasmic (IDHC_DICDI)	1
	cytochrome P450 2D2O (CP2DK_MESAU)	1
	adenylate kinase isoenzyme 5 (KAD5_HUMAN)	1
	cytochrome P450 3A18 (CP3AI_RAT)	1
	ketohexokinase (KHK_HUMAN)	1
	carboxypeptidase M (CBPM_PONAB)	1
	alpha-amylase (AMY_PECMA)	2
microtubule associated proteins	tubulin alpha 2,4 chain (TBA2_PATVU)	1
	tubulin beta chain (TBB_LYTPI)	1
	outer dynein arm-docking complex subunit 3(CC151_BOVIN)	1
	tubulin beta-6 chain (TBB6_ECTVR)	1
	tubulin beta chain (TBB_PARLI)	1
	outer dynein arm-docking complex subunit 4 (TTC25_DANRE)	1
	dynein heavy chain, cytoplasmic (DYHC_EMENI)	1
	actin, muscle (ACTM_LYTPI)	1
	alpha-actinin (ACTN_DERFA)	1
	tubulin beta chain (TBB_STRPU)	1
protein trafficking	ADP-ribosylation factor (ARF_ASHGO)	1

## Data Availability

The data presented in this study are openly available in GenBank, reference number PRJNA825750.

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
