# Peer review of "The Mantle Transcriptome of Chamelea gallina (Mollusca: Bivalvia) and Shell Biomineralization"

_animals, 2022, doi:10.3390/ani12091196_

Round 1

Reviewer 1 Report

This is a welcome paper to the literature on the expression of genes encoding proteins involved in Bivalvia biomineralization process in response to biotic and abiotic factors.

The manuscript is clearly written, terminology meets accepted standards, methods are adequate and well described, and references are exhaustive. The presentation of the results and their discussion are scientifically sound, and the authors made a good job into placing them in an appropriate context. I see significant citation potential in this manuscript and recommend it for publication in its almost current state.

The only problem I see is technical and can easily be fixed: the captions to the drawings are dramatically confused. This needs to be tidied up.

Author Response

Reviewer 1

This is a welcome paper to the literature on the expression of genes encoding proteins involved in Bivalvia biomineralization process in response to biotic and abiotic factors.

The manuscript is clearly written, terminology meets accepted standards, methods are adequate and well described, and references are exhaustive. The presentation of the results and their discussion are scientifically sound, and the authors made a good job into placing them in an appropriate context. I see significant citation potential in this manuscript and recommend it for publication in its almost current state.

The only problem I see is technical and can easily be fixed: the captions to the drawings are dramatically confused. This needs to be tidied up.

Answer: We thank the anonymous reviewer for his/her comments. We addressed his/her request and solved the problem.

Reviewer 2 Report

Manuscript titled " The mantle transcriptome of Chamelea gallina (Mollusca: Bivalvia) and shell biomineralization" (Animals 1679898) by Carducci et al. aims the ability of Chamelea gallina to modulate the expression of genes encoding proteins involved in shell biomineralization process in response to abiotic and biotic factors. To carry out this work, authors provided the first comprehensive transcriptome from mantle, the tissue responsible for shell formation.

In general, this work is of interest for better understand the function of differentially expressed genes that play an essential role related to biomineralization process.

It is also a good reference work to researchers working with other mollusc species.

Study is original and fairly well structured. Results are clearly presented and sufficiently discussed. Its content in general justifies the length. Language is clear and understandable. Figures have good quality and well-illustrated.

However, in my opinion some paragraphs  should be improved and I suggest some shortenings below.

Throughout the text, except in the summary paragraph, the authors do not explain why biomineralization is important. Moreover, although biochemical or biological processes have been introduced and specified, a reader unfamiliar with the subject, reading the sentence in L305-307, might infer that biomineralization is a problem. In my opinion, a sentence with a little explanation should be complemented in the Introduction and in Discussion.

L99-100, L166, L173,  L200-202 It is not clear in these sentences on the basis of which criteria the pools were created. I mean, the subsets and the 10 specimens of 30 were chosen randomly or in relation to morphological characteristics, shells, size ... etc? In addition, in L274-277 authors stated that clams sampled at S site displayed a more homogeneous organization of the shell matrix than SM individuals, but is it in relation to the only selected pools or to all individuals sampled in those areas?

In Figure 4 please indicate the site station in relation to the colour of the lines.

Author Response

Reviewer 2

Manuscript titled " The mantle transcriptome of Chamelea gallina (Mollusca: Bivalvia) and shell biomineralization" (Animals 1679898) by Carducci et al. aims the ability of Chamelea gallina to modulate the expression of genes encoding proteins involved in shell biomineralization process in response to abiotic and biotic factors. To carry out this work, authors provided the first comprehensive transcriptome from mantle, the tissue responsible for shell formation.

In general, this work is of interest for better understand the function of differentially expressed genes that play an essential role related to biomineralization process.

It is also a good reference work to researchers working with other mollusc species.

Study is original and fairly well structured. Results are clearly presented and sufficiently discussed. Its content in general justifies the length. Language is clear and understandable. Figures have good quality and well-illustrated.

However, in my opinion some paragraphs should be improved and I suggest some shortenings below.

Throughout the text, except in the summary paragraph, the authors do not explain why biomineralization is important. Moreover, although biochemical or biological processes have been introduced and specified, a reader unfamiliar with the subject, reading the sentence in L305-307, might infer that biomineralization is a problem. In my opinion, a sentence with a little explanation should be complemented in the Introduction and in Discussion.

Answer: We thank the reviewer for his/her comments that helped us to improve the manuscript. Following his/her suggestions, we added a sentence to explain the importance of biomineralization in the Introduction and Discussion sections.

L99-100, L166, L173, L200-202 It is not clear in these sentences on the basis of which criteria the pools were created. I mean, the subsets and the 10 specimens of 30 were chosen randomly or in relation to morphological characteristics, shells, size ... etc?

Answer: We clarified this aspect in the main text following the reviewer suggestions.

In addition, in L274-277 authors stated that clams sampled at S site displayed a more homogeneous organization of the shell matrix than SM individuals, but is it in relation to the only selected pools or to all individuals sampled in those areas?

Answer: we thank the reviewer for this comment. We modified the material and methods section explaining that for SEM observations “…For each sampling site, SEM observations were carried out on a subset of individuals from those selected for RNA-Seq analyses….” and thus we believe that now the misunderstanding has been solved.

In Figure 4 please indicate the site station in relation to the colour of the lines.

Answer: done.

Reviewer 3 Report

An excessive misunderstanding of the biomineralization process prevents valuable interpretation of your results

Topic: In two distinct sites of the Adriatic seashore, samples from populations  of Chamelea gallina (Pelecypod Mollusc) where collected in order to investigate  whether the animals are able tomodulate the expression of genes encoding proteins involved in biomineralization process in response to biotic and abiotic factors.

Methods   Mantle activity in shell formation was expressed by RNA sequencing with functional annotations.  Shell mineralogy was characterize by Xray diffraction and shell microstructures described by SEM observation of transversal polished and etched sections.

Environmental characterization was obtained by time series measurements for salinity and temperatures.

Results   Discussion chapter comprises considerations about the potential role of some genes  with conclusion that food availability and salinity might influence shell formation in C. gallina.

Comment:  From a methodological view point remark must be made that gene expressions were obtained by using the shell mantle as a whole, without consideration to the distinct mineralization areas that are simultaneously active in the shell construction process.

               This point is surprising because authors are aware of the existence of distinct microstructures in the Chamelea shells (introduction , line 37 to 43).  It is long established that each distinct shell layer is built by a specific panel of organic compounds  (even if their respective roles are far from being established). Therefore to obtain a valid relationship between gene expressions and mineralization, comparison must be conducted between the microstructures  visible on the internal shell surfaces and the corresponding areas of the mantle. The distinct mineralizing area of the mantle produce the microstructurally distinct  and concentrically arranged zones of the internal surface (facing the mantle areas).

This is exactly what is done in production of cultivated pearls, for instance.  Pearl are made of nacre (an aragonite form that is specific to some mollusk families, and more specifically in the Pterioid Pelecypods). In order to get nacreous pearls, the pearl producers cut a small ribbon of the nacre producing tissues from the mantle (nacre in the internal layer of the mantle), avoiding to include fragments of the concentric outer layer (that is producing calcite prisms in this family).      

This specialized subdivision of the mollusk mantle with respect to biomineralization is a common feature throughout the whole phylum. The shell layers are built by superposition of the micrometer thick growth layers produced by mantle during the dimensional expansion of the shell. The specialized mineralizing areas of the mantle are moving during this expansion of the shell, resulting in the microstructural sequence of the shell layers.     

Conclusion is clear that using the mantle as a whole, as did the authors of this manuscript, results in a confusing gene expression that prevent relevant comparison to be made, specifically with respect to “thickness and sturdiness of the shells”.

Getting significant results concerning this interesting topic requires a better understanding of the biomineralization process through time, as did the Japanese pearl producers in the beginning of le late century (in spite of their complete ignorance of genetics for historical reason). The concentrical mineralizing areas of the mantle must be separately treated and their respective gene expressions compared to shell microstructural patterns in the corresponding areas.

Author Response

Reviewer 3

An excessive misunderstanding of the biomineralization process prevents valuable interpretation of your results.

Topic: In two distinct sites of the Adriatic seashore, samples from populations of Chamelea gallina (Pelecypod Mollusc) where collected in order to investigate whether the animals are able to modulate the expression of genes encoding proteins involved in biomineralization process in response to biotic and abiotic factors.

Methods   Mantle activity in shell formation was expressed by RNA sequencing with functional annotations.  Shell mineralogy was characterized by Xray diffraction and shell microstructures described by SEM observation of transversal polished and etched sections.

Environmental characterization was obtained by time series measurements for salinity and temperatures.

Results   Discussion chapter comprises considerations about the potential role of some genes with conclusion that food availability and salinity might influence shell formation in C. gallina.

Comment:  From a methodological view point remark must be made that gene expressions were obtained by using the shell mantle as a whole, without consideration to the distinct mineralization areas that are simultaneously active in the shell construction process.

               This point is surprising because authors are aware of the existence of distinct microstructures in the Chamelea shells (introduction, line 37 to 43).  It is long established that each distinct shell layer is built by a specific panel of organic compounds (even if their respective roles are far from being established). Therefore to obtain a valid relationship between gene expressions and mineralization, comparison must be conducted between the microstructures visible on the internal shell surfaces and the corresponding areas of the mantle. The distinct mineralizing area of the mantle produce the microstructurally distinct  and concentrically arranged zones of the internal surface (facing the mantle areas).

This is exactly what is done in production of cultivated pearls, for instance.  Pearl are made of nacre (an aragonite form that is specific to some mollusk families, and more specifically in the Pterioid Pelecypods). In order to get nacreous pearls, the pearl producers cut a small ribbon of the nacre producing tissues from the mantle (nacre in the internal layer of the mantle), avoiding to include fragments of the concentric outer layer (that is producing calcite prisms in this family).      

This specialized subdivision of the mollusk mantle with respect to biomineralization is a common feature throughout the whole phylum. The shell layers are built by superposition of the micrometer thick growth layers produced by mantle during the dimensional expansion of the shell. The specialized mineralizing areas of the mantle are moving during this expansion of the shell, resulting in the microstructural sequence of the shell layers.     

Conclusion is clear that using the mantle as a whole, as did the authors of this manuscript, results in a confusing gene expression that prevent relevant comparison to be made, specifically with respect to “thickness and sturdiness of the shells”.

Getting significant results concerning this interesting topic requires a better understanding of the biomineralization process through time, as did the Japanese pearl producers in the beginning of le late century (in spite of their complete ignorance of genetics for historical reason). The concentrical mineralizing areas of the mantle must be separately treated and their respective gene expressions compared to shell microstructural patterns in the corresponding areas.

Answer: We really thank the reviewer for his/her comments. The aim of our work was to investigate for the first time the biomineralization process in Chamelea gallina providing a general overview on genes involved in shell formation in this bivalve. We agree with the reviewer that an analysis on distinct mineralization mantle areas will provide more detailed information about the shell formation process. However, this experimental approach would constitute new and different research that could be considered in the future to implement our results.

Round 2

Reviewer 3 Report

Rapport 2

Authors’ Answer:

We really thank the reviewer for his/her comments. The aim of our work was to investigate for the first time the biomineralization process in Chamelea gallina providing a general overview on genes involved in shell formation in this bivalve. We agree with the reviewer that an analysis on distinct mineralization mantle areas will provide more detailed information about the shell formation process.

However, this experimental approach would constitute new and different research that could be considered in the future to implement our results.

Reviewer conclusion

Authors recognize that no substantial change has been made to the initial paper regarding the structure of the shells.

 Due to methodological mistake data regarding  shell structure are basically useless as no relationship can be established between the series of genes and the mineralization process.

Therefore this manuscript must be limited to the biochemical part as suggested by authors themselves:  “this experimental approach would constitute new and different research that could be considered in the future to implement our results”.

Author Response

Reviewer conclusion

Authors recognize that no substantial change has been made to the initial paper regarding the structure of the shells.

 Due to methodological mistake data regarding shell structure are basically useless as no relationship can be established between the series of genes and the mineralization process.

Therefore, this manuscript must be limited to the biochemical part as suggested by authors themselves: “this experimental approach would constitute new and different research that could be considered in the future to implement our results”.

Answer: We do not share suggestions made by the reviewer 3. We are sure that our approach is correct as also underlined by the other two reviewers and it is in line with other papers. Here you can find other works in which a similar approach was applied:

  • Scientific Reports, Song et al. 2022 https://www.nature.com/articles/s41598-022-08610-5.pdf
  • Plos One, Jiang et al. 2020 https://journals.plos.org/plosone/article?id=10.1371/journal.pone.0231414
  • Scientific Reports, Li et al. 2016 https://www.nature.com/articles/srep18943
  • Marine Genomics, Sleight & Clark, 2016 https://www.sciencedirect.com/science/article/pii/S1874778716300034
  • Scientific Reports, Zheng et al. 2015 https://www.nature.com/articles/srep14408.

In none of these papers analyses on distinct mineralization mantle areas were performed. We have adopted an integrated and multidisciplinary approach that helps in supporting and discussing our data that result to be solid and provide for the first-time information in C. gallina about genes known to be involved in bivalve shell formation.